

# Upregulated expression of pyruvate kinase M2 mRNA predicts poor prognosis in lung adenocarcinoma

Guiping Wang[1,*], Yingying Zhong[2,*], Jiecong Liang[3], Zhibin Li[1] and Yun Ye[2]

[1] Department of Pharmacy, Guangzhou Health Science College, Guangzhou, China
[2] College of Biological and Chemical Engineering, Guangxi University of Science and Technology, Liuzhou, China
[3] Department of General Surgery, Guangzhou Women and Children Medical Center, Guangzhou, China
* These authors contributed equally to this work.

Corresponding authors
Guiping Wang,
docgpwang@163.com
Yun Ye, yunye@gxust.edu.cn

## ABSTRACT

**Background:** Pyruvate kinase M2 (PKM2) is critical regulator contributing to Warburg effect. However, the expression pattern and prognostic value of PKM2 remain unknown in lung adenocarcinoma (LUAD). The aim of this study is to clarify the prognostic value of PKM2 via intergrated bioinformatics analysis.
**Methods:** Firstly, mRNA expression levels of PKM2 in LUAD were systematically analyzed using the ONCOMINE and TCGA databases. Then, the association between PKM2 expression and clinical parameters was investigated by UALCAN. The Kaplan–Meier Plotter was used to assess the prognostic significance of PKM2. Finally, the relationship between PKM2 expression and its genetic and epigenetic changes was evaluated with MEXPRESS and MethHC database.
**Results:** Pooled analysis showed that PKM2 is frequently upregulated expression in LUAD. Subsequently, PKM2 expression was identified to be positively associated with tumor stage and lymph node metastasis and also strongly correlated with worse OS ($P = 2.80e-14$), PPS ($P = 0.022$), FP ($P = 1.30e-6$) and RFS ($P = 3.41e-8$). Importantly, our results demonstrated that over-expressed PKM2 is associated with PKM2 hypomethylation and copy number variations (CNVs).
**Conclusion:** This study confirms that over-expressed PKM2 in LUAD is associated with poor prognosis, suggesting that PKM2 might act as a promising prognostic biomarker and novel therapeutic target for LUAD.

## INTRODUCTION

Lung adenocarcinoma (LUAD), one of the most common malignant tumors worldwide, accounts for nearly 40∼50% of primary lung cancers. In recent decades, although the predominant improvement in the early-diagnosis, surgical resection and molecular targeted drugs, the prognosis of patients with LUAD is still extremely poor because of late-diagnosis and metastasis (*Saito et al., 2016*, *2018*). Therefore, it is essential to identify specific and sensitive molecular biomarkers for improving the diagnosis and prognosis of patients with LUAD.

Energy metabolism of cancer has received striking attention in last several decades. "The Warburg effect" is regarded as an important hallmark of cancer cells (*Vander Heiden, Cantley & Thompson, 2009*). Most cancer cells prefer to produce the energy through aerobic glycolysis instead of oxidative phosphorylation even in the presence of sufficient oxygen. Recently, a number of evidents have also implicated that "reversing Warburg effect" might provide a novel strategy for cancer diagnosis and treatment (*Poteet et al., 2018*). Thus, gaining insight into the important regulated moleculars contributing to "Warburg effect" might facilitate the identification of novel and potential prognositic biomarkers.

Pyruvate kinase (PKM2), a key metabolic enzyme for the last rate-limiting step of glycolysis, catalyzes the transfer of a phosphoryl group from phosphoenolpyruvate to ADP, generating ATP and pyruvate. The emerging studies have suggested that PKM2 plays an important role in cancer metabolism, invasion, metastasis and chemoresistance (*Dayton, Jacks & Vander Heiden, 2016*). Given that PKM2 is an important driver regulating energy metabolism of cancer cells, targeting PKM2 might be an promising therapeutic strategy. Recent studies have reported that high PKM2 expression is associated with an unfavorable prognosis in multiple human tumors including hepatocellular carcinoma (*Lv et al., 2018*), breast cancer (*Yang et al., 2017*) and osteosarcoma (*Liu et al., 2016*), etc. Nevertheless, over-expression of PKM2 in gastric cancer and pancreatic cancer were not correlated with poor prognosis (*Zhu et al., 2017*). A few studies also have investigated the correlation between the expression of PKM2 and prognosis in lung cancer. For example, over-expression of PKM2 was reported to be related to poor prognosis in LUAD, and knockdown of PKM2 suppressed tumor growth and invasion (*Sun et al., 2015*). In addition, PKM2 expression also was demonstrated to be a predictive biomarker of chemotherapeutic sensitivity, such as platinum (*Papadaki et al., 2014*) and docetaxel (*Yuan et al., 2016*), as well as radiosensitivity in lung cancers (*Meng et al., 2015*). However, another study indicated that PKM2 has no value as predictive markers of NSCLC regardless of the histological type and grade of malignancy (*Kobierzycki et al., 2014*). Up to the date, the prognostic value of PKM2 expression in LUAD remains unclear and inconsistent. Therefore, it is necessary to systematically clarify the prognostic values of PKM2 in LUAD. Additionally, to our knowledge the relationship between PKM2 expression and its genetic and epigenetic changes has not been previously evaluated.

In the present study, we set out to elucidate the prognositic value of PKM2 in LUAD based on intergrated analysis. Firstly, we systematically evaluated PKM2 expression using ONCOMINE database and TCGA data. Subsequently, the association between PKM2 expression and clinical parameters, as well as prognostic values, were investigated. Finally, we explored whether PKM2 expression is related with the changes of PKM2 methylation and copy number variations (CNVs).

## MATERIALS AND METHODS

### ONCOMINE database

ONCOMINE (http://www.oncomine.org) (*Rhodes et al., 2004*), an online microarray database for mining cancer gene information, was used to compare the mRNA levels of

PKM2 between tumor and corresponding normal tissues in different types of cancer. Seven datasets, including Beer Lung (*Beer et al., 2002*), Selamat lung (*Selamat et al., 2012*), Landi lung (*Landi et al., 2008*), Stearman lung (*Stearman et al., 2005*), Hou lung (*Hou et al., 2010*), Okayama lung (*Okayama et al., 2012*) and Su lung (*Su et al., 2007*), were used to identify the expression pattern of PKM2 in LUAD. Statistical differences were determined by Student's $t$-test. The main thresholds were as follows: fold change ≥ 1.5; $P$ value ≤ 0.0001; gene rank ≥ top 10%.

## UALCAN database

UALCAN (http://ualcan.path.uab.edu) is a comprehensive, user-friendly, and interactive web resource for analyzing cancer OMICS data (TCGA and MET500) (*Chandrashekar et al., 2017*). UALCAN data portal can aid in the identification of candidate biomarkers for diagnostic, prognostic and therapeutic implications. The correlation between mRNA levels of PKM2 and clinicopathological features was analyzed to determine the prognostic value of PKM2 in patients with LUAD. The UALCAN provided the statistical significance of all results ($P$-values), and $P$-values < 0.05 were considered statistically significant.

## Survival analysis

Kaplan–Meier Plotter database (http://kmplot.com/analysis/) contains survival information for breast ($n = 6,234$), ovarian ($n = 2,190$), lung ($n = 3,452$), and gastric ($n = 1,440$) cancer patients (*Gyorffy et al., 2013*). Here, the prognostic value of PKM2 in patients with LUAD was verified using Kaplan–Meier survival curve. Patients with LUAD were split into high and low groups based on the best cutoff expression value of PKM2 (Table S1). Kaplan–Meier plots for overall survival (OS), first progression (FP) and post-progression survival (PPS), were drawn. In addition, the prognostic value of PKM2 was further to be assessed using PrognoScan database, which provides a powerful platform for evaluating potential tumor markers and therapeutic targets (*Mizuno et al., 2009*). The hazard ratio with a 95% confidence interval and log rank $P$-value were calculated to evaluate the survival difference between high and low expression groups. $P < 0.05$ was considered statistically significant.

## Methylation and copynumber variations analysis

MEXPRESS is a web-based and user-friendly tool for the visualization of TCGA gene expression, DNA methylation and clinical data, as well as the relationships between them (*Koch et al., 2015*). Considering the critical effect of methylation and CNVs on the expression of the gene, we evaluated the relationship between PKM2 expression and its methylation as well as CNVs using MEXPRESS. A total of 861 samples for LUAC were analyzed and the samples are ordered by their expression levels of PKM2 (Table S2). In addition, the correlation of PKM2 expression in 497 patients with LUAC and their methylation in the promoter and gene body was furtherly assessed using MethHC (Table S4), a database for human pan-cancer gene expression, methylation and microRNA expression (*Huang et al., 2015*). The correlation between PKM2 expression and its
methylation as well as CNVs were confirmed by the Pearson correlation coefficients.
$P < 0.05$ was considered statistically significant.

# RESULTS

## Transcriptional expression levels of PKM2 in LUAC

To address the mRNA expression of PKM2 in multiple human cancers, the expression of
PKM2 in 20 different cancer types were assessed with ONCOMINE database. As shown in
Fig. 1A, mRNA expression of PKM2 were significantly up-regulated in most human
cancer patients except for brain cancer, cervical cancer, esophageal cancer and prostate
cancer. Subsequently, we systematically analyzed mRNA expression levels of PKM2 in
LUAD. As shown in Table 1, our finding revealed that mRNA levels of PKM2 were
significantly higher in LUAC patients with a fold change of 1.6~2.5. By the comparison of
PKM2 expression across seven datasets, pooled analysis also demonstrated that PKM2
was over-expressed in LUAC (Fig. 1B; $P = 1.11e{-}06$). Similar results also were confirmed
from TCGA data (Fig. 1C, $P = 1.624e{-}12$). Collectively, our finding suggests that PKM2
might play an important role in the development of LUAC.

## Relationship between PKM2 expression and clinical parameters in LUAC

Subsequently, we explored the relationship between PKM2 expression and
clinicopathological features, such as tumor stages, lymph node metastasis, smoking,
gender, race and age in patients with LUAC. As shown in Fig. 2, the results revealed that
PKM2 expression was found to be positive associated with tumor stage. The mRNA levels
of PKM2 in patients with stage II and III were apparently higher than that in patients
with stage I (Fig. 2A). Similarly, over-expressed PKM2 was also identified to be positively
correlated with lymph node metastasis (Fig. 2B). Additionally, the expression of PKM2
was observed to significantly elevate in patients(age 21~40 years) (Fig. 2E). However, there
was no relationship between PKM2 expression and other clinical features such as race
(Fig. 2C), smoking (Fig. 2D) and gender (Fig. 2F). Taken together, our results indicated
that increased PKM2 might predict a poor prognosis for patients with LUAC.

## Prognostic value of PKM2 in LUAC

To further evaluate the prognostic value of PKM2, the relationship between mRNA
expression of PKM2 and clinical outcome was assessed using Kaplan–Meier plotter.
As shown in Figs. 3A–3C, Kaplan–Meier plot revealed that over-expressed PKM2 was
significantly associated with worse OS (HR = 2.56; 95% CI [1.99–3.28]; $P = 2.8e{-}14$), PPS
(HR = 1.73; 95% CI [1.08–2.79]; $P = 0.022$) and FP (HR = 2.18; 95% CI [1.58–3];
$P = 1.3e{-}6$), median survival time and other survival information were presented in
Table S3. The TNM staging is still one of powerful survival predictive factors, therefore, we
evaluated the relationship between mRNA levels of PKM2 and survival in the same staging
state. The results also indicated that over-expressed PKM2 was significantly associated
with worse OS in the same staging state (Figs. 3D–3F; Table S3). In addition, similar results
also were identified from PrognoScan database analysis (Table 2). Of interestingly,

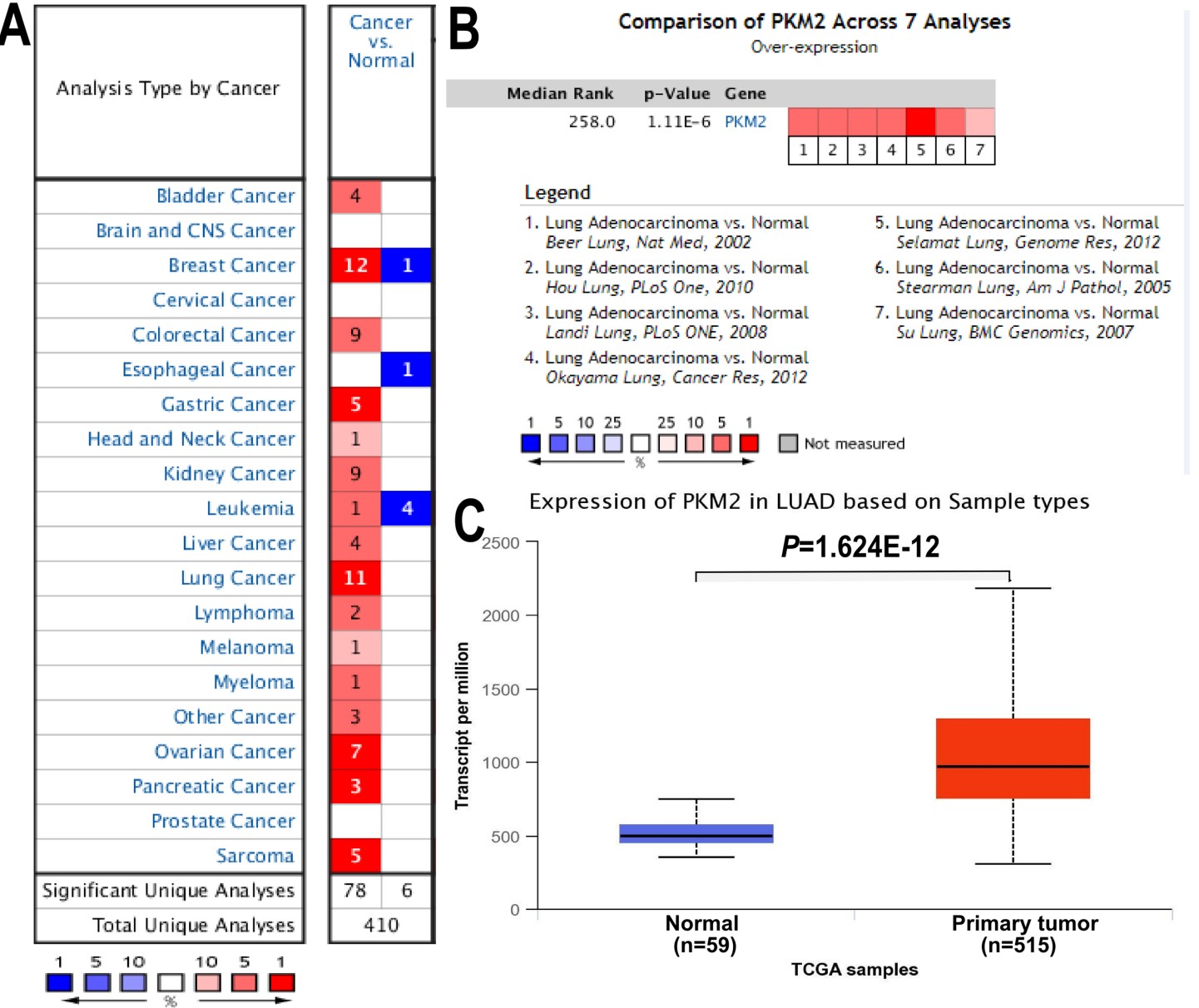

**Figure 1 mRNA expression of PKM2 in LUAC.** The expression of PKM2 in 20 different cancers was evaluated using ONCOMINE database. (A) Red represents up-regulated expression, and blue color for down-regulated expression. Cell color is determined by the best gene rank percentile for the analyses within the cell. (B) The pooled comparison of PKM2 expression across seven analysis in LUAC. The rank for a gene is the median rank for that gene across each of the analyses. The P-value for a gene is its P-value for the median-ranked analysis. (C) Box plot showing mRNA levels of PKM2 in LUAC and normal control from TCGA database. TCGA, The Cancer Genome Atlas.

over-expressed PKM2 was indicated to be associated with significantly shorter RFS (HR = 3.50; 95% CI [2.24–5.45]; P = 3.41e–08). Collectively, our finding suggested that over-expressed PKM2 were significantly associated with poor prognosis in LUAC.

## Effect of methylation and CNVs on PKM2 expression in LUAC

DNA methylation is major epigenetic mechanisms that modulate gene expression and also is an important biological mechanism for tumor occur and development. Here, we

**Table 1 Significant changes of PKM2 expression in LUAC using the ONCOMINE database.**

| No | Datasets | Cases | Fold change | t-test | P-value |
|---|---|---|---|---|---|
| 1 | Beer lung | 96 | 1.783 | 8.244 | 1.23e−07 |
| 2 | Selamat lung | 116 | 2.551 | 12.037 | 3.56e−20 |
| 3 | Landi lung | 107 | 1.763 | 9.753 | 1.22e−16 |
| 4 | Stearman lung | 39 | 1.643 | 6.200 | 1.11e−06 |
| 5 | Hou lung | 156 | 1.832 | 7.565 | 1.55e−11 |
| 6 | Okayama lung | 246 | 1.876 | 8.828 | 9.76e−12 |
| 7 | Su lung | 66 | 1.682 | 3.535 | 6.14e−04 |

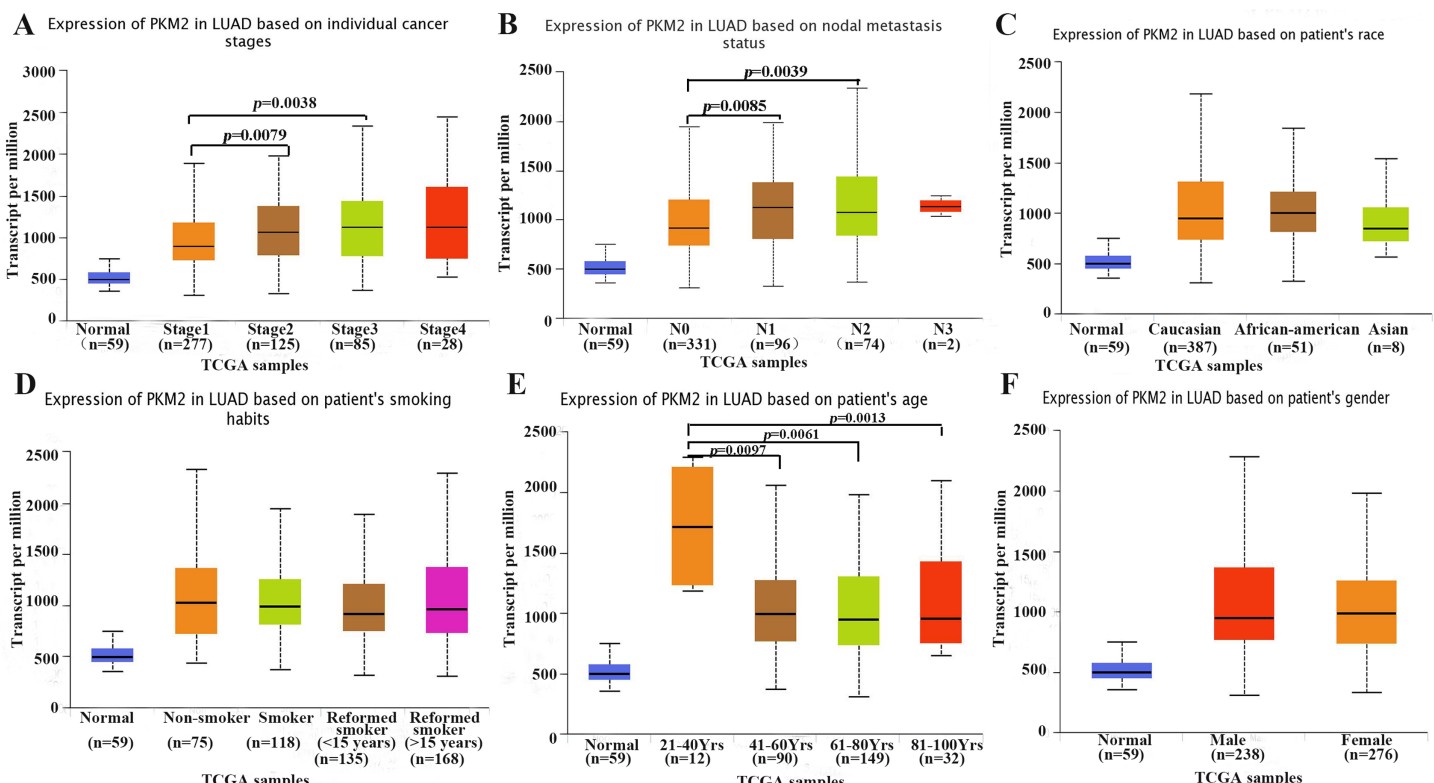

**Figure 2 Box plots showing expression of PKM2 mRNA in LUAC based on clinical parameters.** The UALCAN database was used to assess the relationship between PKM2 expression and clinical parameters such as cancer stages (A), lymph node metastasis status (B), race (C), smoking habits (D), age (E) and gender (F). LUAC, lung adenocarcinoma; TCGA, The Cancer Genome Atlas.

investigated whether PKM2 mRNA expression was related to its methylation using MEXPRESS. As shown in Fig. 4A, the samples are ordered by PKM2 expression, and the plot shows that there is a negative correlation between PKM2 expression and methylation around CpG island and promoter region. Nine abnormal methylation sites (cg19687230, $r = -0.131$; cg00635674, $r = -0.116$; cg27032813, $r = -0.164$; cg07365018, $r = -0.164$; cg22234930, $r = -0.385$; g24327132, $r = -0.346$; cg05888487, $r = -0.138$; cg23160336, $r = -0.112$; cg09651136, $r = -0.224$) were observed and the Pearson correlation coefficients ($-0.116$ to $-0.385$ around the promoter region) was indicated on

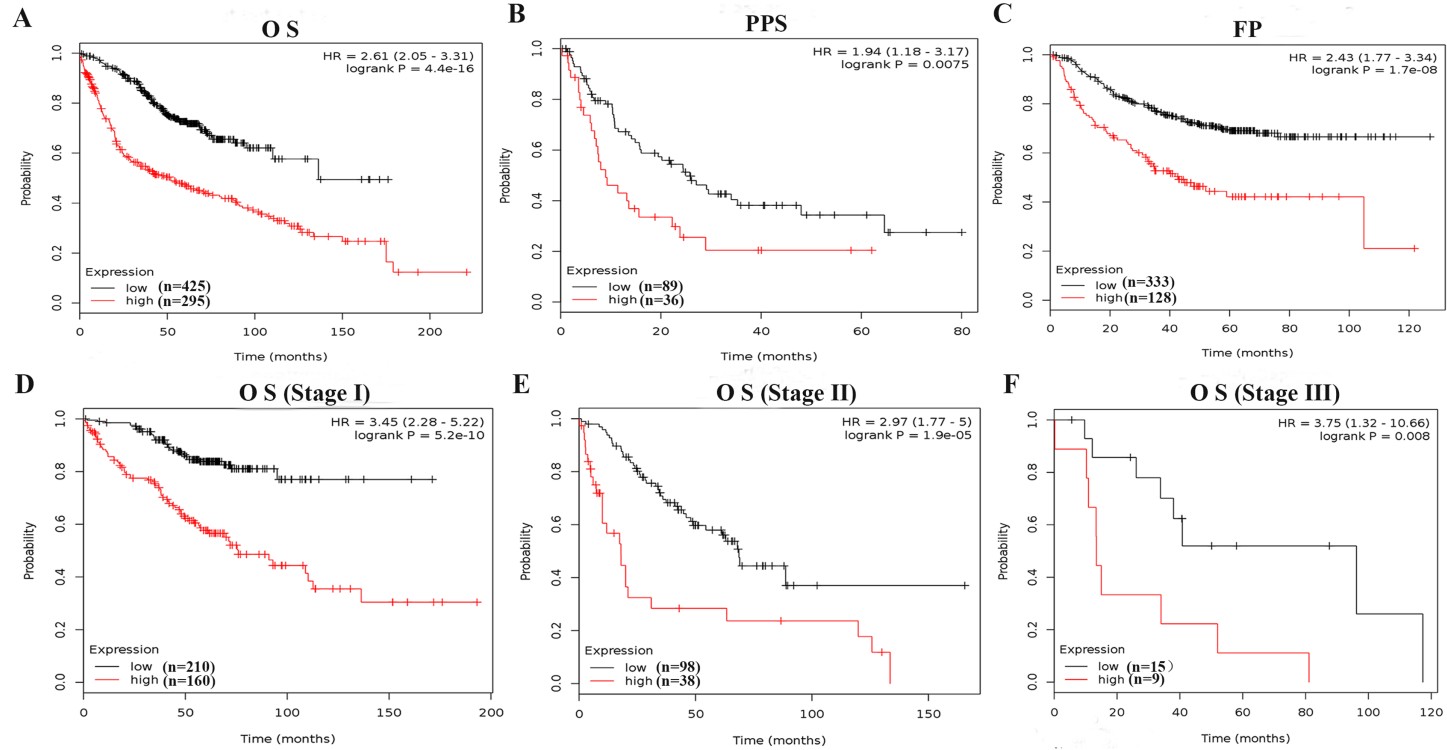

**Figure 3 Survive curves evaluating the prognostic value of PKM2 mRNA expression in LUAC.** Patients with LUAD were split into high and low groups based on the best cutoff expression value of PKM2, OS (A) PPS (B) and FP (C) curves were generated using the Kaplan–Meier Plotter. In addition, the relationship between mRNA levels of PKM2 and survival was furtherly evalvated in stage I (D), stage II (E) and stage III (F), respectively. *P*-values were calculated by log-rank test. HRs with 95% CIs are displayed. OS, overall survival; FP, first progression; PPS, post-progression survival.                                              

**Table 2 Relationship between PKM2 expression and survival trends using PrognoScan database in LUAC.**

| Dataset | Endpoint | Probe ID | Cases | Cox *p*-value | HR |
|---|---|---|---|---|---|
| HARVARD-LC | OS | 32378_at | 84 | 0.0454 | 2.15 (1.02~4.55) |
| Jacob-00182-MSK | OS | 201251_at | 104 | 0.0084 | 4.16 (1.44~12.04) |
| GSE31210 | OS | 201251_at | 204 | 1.30e−6 | 4.56 (2.47~8.44) |
| GSE31210 | RFS | 201251_at | 204 | 3.41e−8 | 3.50 (2.24~5.45) |

**Note:**
OS, overall survival; RFS, relapse free survival.

the right hand side. Meanwhile, MethHC plot also revealed that PKM2 expression was negatively associated with its DNA methylation in the promoter (Fig. 4B, $r = -0.171$, $P < 0.05$), as well as gene body region (Fig. 4C, $r = -0.280$, $P < 0.05$). In addition, MEXPRESS plot also reveals that PKM2 expression is correlated with tumor stage ($P = 0.002$) and OS event status ($r = 0.131$, $P < 0.01$), which was consistent with our above results (Figs. 2 and 3).

Meanwhile, the relationship between PKM2 expression in 861 patients with LUAC and its CNVs was assessed using MEXPRESS. Our finding indicated that there is a strongly

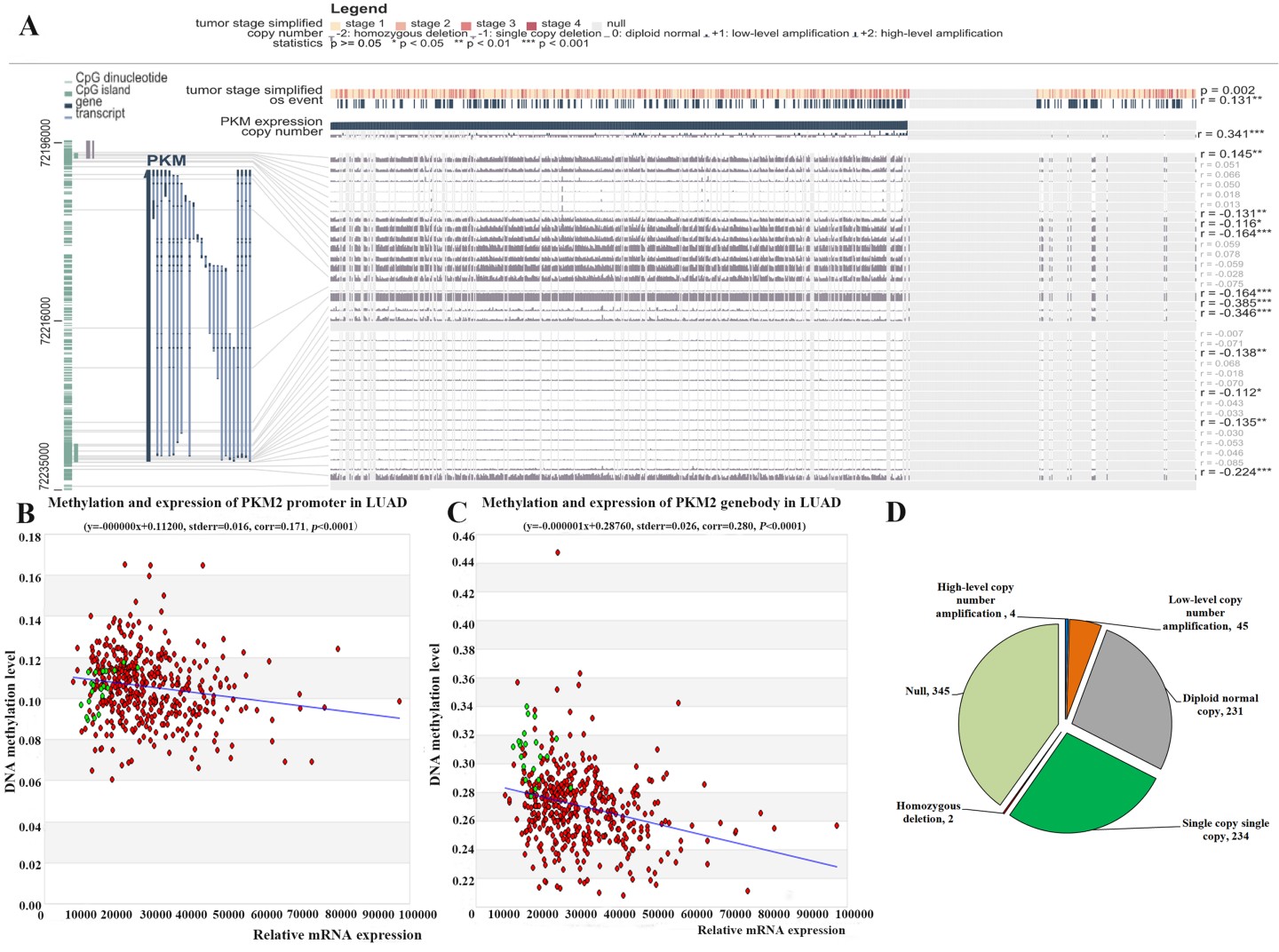

**Figure 4 Relationship between PKM2 expression and its methylation and CNVs in LUAC.** Visualization of TCGA data for PKM2 expression in 861 patients using MEXPRESS (A). The samples are ordered by their expression value. This view shows the relationship between PKM2 expression and methylation around CpG island and promoter region, clinical features, as well as CNVs. Statistical significance was indicated in the right side. The relationship between PKM2 expression and its methylation levels in promoter (B) and genebody (C) was analyzed with MethHC database. The Pie char was plotted for showing PKM2 CNVs changes in LUAC samples (D). CNVs was estimated using the GISTIC2 method. Values: −2, homozygous deletion; −1, single copy deletion; 0, diploid normal copy; 1, low-level copy number amplification; 2, high-level copy number amplification.

positive correlation between PKM2 expression and its CNVs changes (Fig. 4A, r = 0.341, P < 0.0001). Our results subsequently indicated that CNVs of PKM2 frequently occurred in patients with LUAC. As shown in Fig. 4D, there were 49 (5.69%) of 861 patients with copy number of gain (CNG), including 45 samples for low-level copy number amplification, and four samples for high-level copy number amplification; copy number loss (CNL) was frequently detected in 236 (27.41%) of 861 patients, including 2 samples for homozygous deletion and 234 samples for single copy deletion.

## DISCUSSION

According to the Warburg effect, most cancer cells produce energy by glycolysis, suggesting that targeting cancer metabolism might be a potential field in drug discovery (*Liu et al., 2016*; *Lv et al., 2018*; *Yang et al., 2017*). As a key regulator involving in Warburg effect, PKM2 is frequently expressed at high levels in numerous human tumors and play an important role in tumorigenesis, aerobic glycolysis and chemoresistance (*Dayton, Jacks & Vander Heiden, 2016*; *Guo et al., 2019*; *Liu et al., 2016*; *Lv et al., 2018*; *Zhu et al., 2016*). Recent evidents (*Chu et al., 2015*; *Guo et al., 2019*; *Sun et al., 2015*) have indicated that PKM2 is highly over-expressed in LUAC compared to normal lung tissues. Moreover, elevated expression of PKM2 is involved in cell proliferation and tumourigenesis, as well as the regulation of immune and inflammatory responses. More importantly, accumulating evidence has suggested that enhanced PKM2 expression correlates with therapeutic resistance in LUAC (*Meng et al., 2015*; *Yuan et al., 2016*). Furthermore, selective inhibition or knockdown of PKM2 with RNAi leads to suppression of cell proliferation, induction of apoptosis and increase of the sensitivity of tumor cells to chemotherapy in LUAC, suggesting that selective targeting of PKM2 may serve as a potential therapeutic target for LUAD, especially for patients with chemotherapeutic resistance (*Chu et al., 2015*; *Sun et al., 2015*; *Suzuki et al., 2019*; *Wang et al., 2019*; *Yuan et al., 2016*).

At present, a limited number of studies have investigated the prognostic significance of PKM2 in LUAD. The previous evidence indicated that high PKM2 expression was significantly associated with lymph node metastasis and distant metastasis, as well as poor prognosis in LUAD (*Sun et al., 2015*). But unfortunately, the study showed several limitations, for example, the sample size was relatively small, and multivariate analysis of overall survival was not performed. Another study demonstrated that PKM2 was significantly positively associated with PD-L1 expression, as well as over-expression of PKM2 and PD-L1 were related with worse overall survival and disease-free survival in LUAD (*Guo et al., 2019*). However, *Rzechonek et al. (2017)* suggested that PKM2 might not be an ideal diagnostic and prognostic biomarker in NSCLC because the specificity of PKM2 as a cancer marker was 50% for both adenocarcinoma and squamous cell carcinoma. We think that one of main limitations in their study is small sample size with only 45 NSCLC patients, which prevented the data from drawing scientific conclusions. To further elucidate the prognostic value of PKM2 for LUAC, in the present study, we systematically analyzed the expression of PKM2 in LUAC using the ONCOMINE and TCGA database, our result confirmed that PKM2 was over-expressed in patients with LUAC (Fig. 1; Table 1). Subsequently, we evaluated whether over-expression of PKM2 was associated with clinicopathological features and survival outcomes for LUAC patients. In agreement with a previous study (*Sun et al., 2015*), our current finding revealed that high-expression of PKM2 was positively correlated with tumor stage and lymph node metastasis in LUAC (Fig. 2). Meanwhile, the survival analysis furtherly confirmed that over-expressed PKM2 was significantly associated with worse OS, FP, PPS and RFS (Fig. 3), which was consistent with previous results (*Guo et al., 2019*). Moreover, our data

also revealed that higher expression of PKM2 also significantly was associated with shorter OS in the same staging state (Figs. 3D–3F; Table S3). Overall, our current evidences further support the role of PKM2 expression as a prognostic biomarkers and therapeutic targets for LUAC.

Accumulating evidence has demonstrated that epigenetic alterations is important for the development of cancer. DNA methylation, typically occurring in CpG dinucleotides, is an important epigenetic modification of gene transcription, and is strongly related with tumorigenesis by repressing the expression of the tumor suppressor gene and promoting the expression of oncogenes. Thus, the discovery of DNA methylation biomarkers might be a promising approach to improve the early diagnosis. Currently, some DNA methylation signatures have been identified to help improving early detection and predicting the prognosis in LUAC (Wang et al., 2019). Elevated expression of PKM2 has been reported in many cancers, however, only a limited number of studies have investigated the methylation of PKM2 in human cancers (Desai et al., 2014), and the relationship between the expression of PKM2 in LUAC and its methylation is still unknown. Here, our results indicated that PKM2 expression is negatively correlated with its methylation around CpG island and the promoter region (Fig. 4A). Importantly, nine abnormal methylation sites, including cg19687230, cg00635674, cg27032813, cg07365018, cg22234930, g24327132, cg05888487, cg23160336 and cg09651136, were identified to be negatively related with PKM2 expression in LUAC. The similar results were also found in MethHC analysis (Figs. 4B and 4C). MethHC results showed that mRNA expression of PKM2 was correlated with PKM2 hypomethylation in promoter region ($r = 0.171$, $P < 0.0001$, Fig. 4B) and in gene body region ($r = 0.280$, $P < 0.0001$, Fig. 4C). As shown in Table S4, the hypo-methylation of PKM2 in promoter or genebody region was very frequent in LUAC. The percentage of hypo-methylation of PKM2 was 48.51% (211 of 435 patients) in promoter region, among the proportion of the differential fold change (>1.2) was 13.10%. Of great interest, the percentage of hypo-methylation of PKM2 was 94.71% (412 of 435 patients) in gene body region, among the proportion of the differential fold change (>1.2) was 24.60%. Our above finding revealed that hypomethylation of PKM2 in genebody region might be strongly correlated with over-expressed PKM2. However, the effect of PKM2 hypo-methylation in promoter and genebody region on mRNA expression of PKM2 needs to be confirmed in further experiments. Collectively, our results suggested that elevated PKM2 expression was correlated with the hypomethylation status of PKM2 in promoter and genebody region.

Copy number variations (CNVs), including copy number loss (CNL) and copy number gain (CNG), are the variable number of DNA fragments in the human genome. CNVs play an important role in tumorigenesis and cancer progression by regrulating the expression of tumor suppressor genes or oncogenes (Zhao & Zhao, 2016). Many studies reported that the mutations in some cancer-related genes such as EGFR contribute to the tumorigenesis and progression of LUAD, as well as drug resistance (Zheng et al., 2019). Of great importance, CNVs signatures in NSCLCs may offer the possibility of identifying

the origin of tumors whose origin is unknown (*Qiu et al., 2017*). A recent study showed that loss of PKM2 could alter mitochondrial substrate utilization and impairs EC proliferation and migration in vivo (*Stone et al., 2018*). However, little is known about CNVs of PKM2, and the effect of CNVs on PKM2 expression still is not exlpored. In the present study, our result indicated that CNVs of PKM2 frequently occurred in patients with LUAC. In 861 patients with LUAD, CNG of PKM2 was observed in 49 (5.69%) patients, and CNL in 236 (27.41%) patients. Over-expressed PKM2 is strongly correlated with its CNVs (Fig. 4A, $r = 0.341$, $P < 0.0001$), suggesting that the CNVs might be an important regulator for driving the expression of PKM2. Nevertheless, more studies are necessary to determine the relationship between PKM2 expression and its CNVs, as well as mechanisms involved in the regulation of PKM2 expression by CNVs.

Copy number variations (CNVs) and DNA methylation are important genetic and epigenetic modification in tumor, which have been linked to genomic instability and have a substantial impact on gene expression. Recently, the association between tumor mutation burden (TMB) and DNA methylation in non-small cell lung cancers (NSCLCs) was explored, and found that high TMB NSCLCs had more DNA methylation aberrance and copy number variations (CNVs) (*Cai et al., 2019*). In the present study, we also demonstrated that CNVs and DNA hypomethylation most frequently occurs in LUAD, which was consistent with *Cai et al.'s (2019)* research. However, our study only suggests that high PKM2 expression is associated with CNVs and DNA hypomethylation, which still needs to be confirmed in further experimental. In addition, both CNVs and hypomethylation might play an important role in up-regulating PKM2 expression; nevertheless, further researches are needed to investigate which one is more important.

## CONCLUSIONS

In summary, our work systematically investigated the expression of PKM2 in LUAD, and identified that over-expressed PKM2 was significantly associated with increasing tumor stage, lymph node metastasis, and worse survival time (OS, PPS, FP and RFS). Most importantly, this integrated study revealed DNA methylation and CNVs might play an vital role in regulating the expression of PKM2 in LUAD. Our findings suggested PKM2 might be used as a promising prognostic biomarker and potential therapeutic target against LUAD or other tumors. Further experiments are needed to verify PKM2 prognostic value for LUAD, and elucidate the roles of these observed genetic and epigenetic aberrations in LUAD.

### Funding

This work was supported by a grant from the Guangdong Science and Technology Project (2016A020215213), and Scientific Research Project of Guangzhou Education Bureau (1201620370). The funders had no role in study design, data collection and analysis, decision to publish, or preparation of the manuscript.

## Grant Disclosures

The following grant information was disclosed by the authors:
Guangdong Science and Technology Project: 2016A020215213.
Scientific Research Project of Guangzhou Education Bureau: 1201620370.

## Competing Interests

The authors declare that they have no competing interests.

## Author Contributions

- Guiping Wang conceived and designed the experiments, analyzed the data, authored or reviewed drafts of the paper, and approved the final draft.
- Yingying Zhong performed the experiments, prepared figures and/or tables, authored or reviewed drafts of the paper, and approved the final draft.
- Jiecong Liang analyzed the data, prepared figures and/or tables, and approved the final draft.
- Zhibin Li analyzed the data, prepared figures and/or tables, authored or reviewed drafts of the paper, and approved the final draft.
- Yun Ye conceived and designed the experiments, authored or reviewed drafts of the paper, and approved the final draft.

## Data Availability

Raw data are available as Supplemental Files.

## Supplemental Information

Supplemental information for this article can be found online at http://dx.doi.org/10.7717/peerj.8625#supplemental-information.

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
