# Peer review of "Upregulated expression of pyruvate kinase M2 mRNA predicts poor prognosis in lung adenocarcinoma"

_PeerJ, doi:10.7717/peerj.8625_

## Round 0.1 · original submission · Major Revisions

Manuscript entitled "Upregulated expression of pyruvate kinase M2 predicts poor prognosis in lung adenocarcinoma" which you submitted to PeerJ, has been reviewed. The reviewers have recommended publication pending major revisions. Therefore, I invite you to respond to the reviewers' comments at the bottom of this letter and revise your manuscript accordingly.

However, there are still more concerns as follows.

(1) In most databases, such as TCGA and GEO, PKM2 can't be distinguished from PKM1 and other transcript variants of PKM gene. Therefore, the data of PKM2's expression is actually the expression of the whole PKM gene. Please clarify it.

(2) There are a lot of grammatical errors. Please carefully check the manuscript.

Reviewer 1 ·

Basic reporting

The present manuscript provides useful information how PKM2 expression is important in lung cancer as a biomarker of prognosis and pathogenesis. Their skillful analysis of data from all subsets reveal its role in different clinical cohorts. These results are important for clinical oncologists who have been treating the patients with lung cancer.

The manuscript is well designed and written to convey the main message to the scientific community. Introduction, study design and data analysis sections systematically outline the study undertaken. They have discussed their results by performing a comparative assessment adequately. They have also investigated the data whether PKM2 mRNA expression was related to its methylation using MEXPRESS, which is a web-based and user-friendly tool for the visualization of TCGA gene expression to strengthen their conclusion.

Experimental design

In this manuscript, the authors have examined overexpression of PKM2 in lung adenocarcinoma (LUAD) using an online microarray database (ONCOMINE database) for mining PKM2 gene information, which is significantly associated with increasing tumor stage, lymph node metastasis, and the copy number variations, survival time. This study includes seven datasets such as Beer Lung, Selamat lung, Landi lung, Stearman lung, Hou lung, Okayama lung and Su lung to analyze PKM2 expression pattern in
LUAD. Further, the study demonstrates that DNA methylation and copy number variations regulate the expression of PKM2 in LUAD, which suggest that PKM2 can serve as a promising prognostic biomarker and potential therapeutic target against LUAD or other tumors.

Validity of the findings

The manuscript is well designed and written to convey the main message to the scientific community. Introduction, study design and data analysis sections systematically outline the study undertaken. They have discussed their results by performing a comparative assessment adequately. They have also investigated the data whether PKM2 mRNA expression was related to its methylation using MEXPRESS, which is a web-based and user-friendly tool for the visualization of TCGA gene expression to strengthen their conclusion.

Additional comments

The present study outlines the role of key cancer metabolism enzyme PKM2 in the disease pathogenesis and prognosis of lung adenocarcinoma to elucidate the prognostic value of PKM2 in LUAD based on integrated analysis of LUAD samples in different subsets. The study focuses on examining the association between PKM2 expression and clinical parameters, as well as prognostic values. whether PKM2 expression is related with the changes of PKM2 methylation and copy number variations (CNVs).

The authors should address following comments and revise the manuscript.
1. How survival was defined in these subsets? Whether it was the time interval from the date of surgery to the date of death? In order to elucidate a possible correlation between PKM2 gene expression and the clinical outcome, please be selective to include the selected patients with survival>certain days (X number of days), indicating that the patient survived from the initial therapeutic intervention (surgery and radiation treatments).
2. The Materials and Methods section should have a separate detailed description for Statistical Analysis including various statistical tools used in the data analysis. To further delineate the role of PKM2 expression, the authors should provide more details to describe plots of the Kaplan–Meier analysis with appropriate sample size, which may provide the information on the length of survival, median survival time of the distinct sample populations, and significance of the difference between the survival curves.
3. What were the common determinants among patients among seven subsets? The authors should include a paragraph in the Result section to describe Patient Characteristics and analyze the data for conveying more meaningful message.
4. Introduction section (lines 86-90) and Discussion section (lines 213-215) summarize the tumor biology of PKM2, its role as diagnostic and prognostic biomarker of the pathogenesis. But, it lacks recently published information by other investigators for its role as either diagnostic or prognostic marker at subcellular levels in a variety of pre-clinical animal models of human cancers including small cell and non-small cell lung cancer. The authors should run a comparative analysis of their results with these published data for a clearer interpretation.

5. The authors should increase the font size of figure title, X and y-axes legends of Figure 2 A-F, Figure 3 A-C, Figure 4 for better readability.

Reviewer 2 ·

Basic reporting

no comment

Experimental design

no comment

Validity of the findings

no comment

Additional comments

Title:
Upregulated expression of pyruvate kinase M2 predicts poor prognosis in lung adenocarcinoma

Comments:

1. The term of “mRNA” should be added to the title of figure 2 and 3, and modified as following:
Figure 2 Box plots showing expression of PKM2 mRNA in LUAC based on clinical parameters.
Figure 3 Survive curves evaluating the prognostic value of PKM2 mRNA expression in LUAC.

2. Figure 2C: Expression of PKM2 in LUAD based on patient’s race.
Comment:
Most of the patients in this study were the Caucasian (N=387). Therefore, the title of this paper should be modified as: Upregulated expression of pyruvate kinase M2 predicts poor prognosis in the Caucasian lung adenocarcinoma
P.S. Asian, N=8; African-American, N= 51.

3. Figure 3 Survive curves evaluating the prognostic value of PKM2 expression in LUAC.
Question:
3.1 How your determined the mRNA threshold of PKM2 to tell the difference between the high and low expression groups? And what was the rationale?
3.2 We would like to know how many percentage of lung adenocarcinoma patients were with high expression of PKM2 mRNA in this study.

4. Figure 4B The relationship between PKM2 expression and its methylation levels in promoter
Figure 4C The relationship between PKM2 expression and its methylation levels in genebody.
Suggestion:
4.1 The labelling of Y axis should be modified from “methylation” to “DNA methylation level”. For X-axis labelling, please modified from “expression” to “relative mRNA expression of promotor” or “relative mRNA expression of genebody”
4.2 The correlation Coefficients (r) of figure 4B and 4C were 0.171 and 0.280 respectively. Please show the p value of each figure for us.
4.3 The significance of hypo-methylation of PKM2 in genebody region should be explored more extensively in the section of discussion if p value of figure 4C less than 0.05.
4.4 What is percentage of hypo-methylation of PKM2 in promoter or genebody of lung adenocarcinoma patients in your study?

5. Figure 4C The Pie char was plotted for showing PKM2 CNVs changs in LUAC samples. The value of diploid normal copy (0) was 27% (231/861).
5.1 There was a wrong spelling, i.e. changs, please correct it as changes.
5.2 The figures shown in line 253~255 were--, CNG of PKM2 was observed in 49 (5.69%) patients, and CNL in 236 (27.41%) patients. It was not match to the labelling of pie chart as shown in figure 4C. Please clarify the mismatch figures in pie char and text.

6. Line240~241, --, hypomethylation of PKM2 in genebody region might be strongly correlated with overexpressed.
Line 254~255, Over-expressed PKM2 is strongly correlated with its CNVs (Fig.
255 4A, r=0.341, p<0.0001),
Question
Both methylation and CNVs had effect on PKM2 expression of lung adenocarcinoma patients as stated by the authors. We would like to which one is more important? Would you please do a couple of comments in the section of discussion?

7. Did the PKM2 can be detected by immuno-histochemical (IHC) method? What is percentage of high expression of PKM2 protein of lung adenocarcinoma patients in this study?

8. The figure of Inverse correlation between DNA methylation(Y axis) and relative mRNA expression (X axis) of promotor or genebody were shown in Figure 4B and C. Please show another figure with Inverse correlation between DNA methylation(Y axis) and protein expression (X axis), if PKM2 protein could be detected by immune-histochemical method.

9. P166, 166 PKM2 was aslo identified to be positively correlated with lymph node metastasis (Fig. 2B)
Suggestion: wrong spelling, asloalso

10. Figure 2 Box plots showing expression of PKM2 in LUAC based on clinical parameters. The mRNA levels of PKM2 in patients with stage II and III were apparently higher than that in patients with stage I (Fig. 2A).
Over-expressed PKM2 was also identified to be positively correlated with lymph node metastasis (Fig. 2B).
Fig. 3, Kaplan–Meier plot revealed that over-expressed PKM2 was significantly associated with worse OS.
Comment:
The TNM staging is still one of powerful survival predictive factors, therefor, the power of PKM2 should be tested in the same TNM stage, or been analyzed with multi-variable Cox regression method. Please show the survival curve of high and low expression of PKM2 of lung adenocarcinoma patients within the same stage.

11. L82~84, Pyruvate kinase (PKM2), a key metabolic enzyme for the last rate-limiting step of glycolysis, catalyzes the transfer of a phosphoryl group from phosphoenolpyruvate to ADP, generating ATP and pyruvate.
L236~238, Importantly, nine abnormal methylation sites, including cg19687230, cg00635674, cg27032813, 237 cg07365018, cg22234930, g24327132, cg05888487, cg23160336 and cg09651136, were238 identified to be negatively related with PKM2 expression in LUAC.
Comment
The PKM2 was treated as oncogene as authors stated, I just wonder what will be happened if PKM2 were hyper-methylation? Have you ever tried to use Gene Ontology (GO) to clarify what biological process will be altered if PKM2 were hyper-methylation?

---

## Round 0.2 · Minor Revisions

The manuscript entitled "Upregulated expression of pyruvate kinase M2 predicts poor prognosis in lung adenocarcinoma" which you submitted to PeerJ, has been reviewed. The reviewers have recommended publication pending minor revisions. Therefore, I invite you to respond to the reviewers' comments at the bottom of this letter and revise your manuscript accordingly.

Reviewer 1 ·

Basic reporting

The present manuscript focuses on a potential role of PKM2 expression as a biomarker of disease severity and prognosis of lung cancer subjects. Their skillful analysis of data from all subsets reveal its role in different clinical cohorts. These results are important for clinical oncologists who have been treating the patients with lung cancer.

The manuscript is well designed and written to convey the main message to the scientific community. The authors have presented their results adequately and incorporated the changes recommended by the reviewers.

I have observed spelling mistakes at certain places, which I have listed in the section "General Comments for Authors" . I have also noticed that there are few key references missing that are pertinent to the study are not covered yet for a comparative assessment of their data.

Experimental design

The authors have improved the experimental design by incorporating the comments and suggestions made by the reviewers.

I have no additional comments.

Validity of the findings

The quality of revised version of the manuscript is improved and conveys the main message to the scientific community adequately. Introduction, study design and data analysis sections systematically outline the study undertaken. They have also investigated the data whether PKM2 mRNA expression was related to its methylation using MEXPRESS, which is a web-based and user-friendly tool for the visualization of TCGA gene expression to strengthen their conclusion.

I noticed that there is a room to introduce some recent references published by other scientists, which they can discuss on a comparative assessment platform for conveying their outcome precisely.

Additional comments

The revised manuscript is improved for its readability and quality. The authors should fix spelling mistakes still uncorrected in lines 193, 195, 265 and 267. It will be better to get the manuscript be reviewed by a English speaking person for narration and grammatical errors.

The manuscript is still missing a comparative assessment of their results with pertinent key references published in the recent past. The authors should run a literature search with appropriate key words and incorporate in the revised manuscript. This is still missing. A few representative references are listed for their review and placing at appropriate sections in the manuscript. They may find more upon searching in literature search engines.
1. Guo et al. The prognostic value of PKM2 and its correlation with tumour cell PD-L1 in lung adenocarcinoma., BMC Cancer. 2019, 19(1), 289
2. Yang et al. Cytosolic PKM2 stabilizes mutant EGFR protein expression through regulating HSP90-EGFR association. Oncogene 2016;35: 3387-98.
3. Gines et al. Subcellular Localization Is Involved in Oxaliplatin Resistance Acquisition in HT29 Human Colorectal Cancer Cell Lines. PLoS One 2015;10: e0123830.
4. Li et al. Nuclear PKM2 contributes to gefitinib resistance via upregulation of STAT3 activation in colorectal cancer. Sci Rep 2015;5: 16082.
5. Suzuki et al. Subcellular compartmentalization of PKM2 identifies anti-PKM2 therapy response in vitro and in vivo mouse model of human non-small-cell lung cancer. PLoS One, 2019, 14(5), e0217131
6. Stone et al. Loss of pyruvate kinase M2 limits growth and triggers innate immune signaling in endothelial cells Nat Commun. 2018; 9(1): 4077
7. Goldberg MS, Sharp PA. Pyruvate kinase M2-specific siRNA induces apoptosis and tumor regression. J Exp Med 2012;209: 217-24.
8. Li et al. PKM2 and ACVR 1C are prognostic markers for poor prognosis of gallbladder cancer. Clin Transl Oncol 2014;16: 200-7.
9. Zhou et al. Role of isoenzyme M2 of pyruvate kinase in urothelial tumorigenesis. Oncotarget 2016;7: 23947-60.

Reviewer 2 ·

Basic reporting

see below

Experimental design

see below

Validity of the findings

see below

Additional comments

Title:
Upregulated expression of pyruvate kinase M2 predicts poor prognosis in lung adenocarcinoma.

Comments:
1. Title: Upregulated expression of pyruvate kinase M2 predicts poor prognosis in lung adenocarcinoma.
Suggestion:
Upregulated expression of pyruvate kinase M2 mRNA predicts poor prognosis in lung adenocarcinoma.
2. P249-252, Thus, the discovery of DNA methylation biomarkers might be a promising approach to improve the early diagnosis. Currently, some DNA methylation signatures have been identified to help improving early detection and predicting the prognosis in LUAC.
Comment:
From my point of view, for clinical application, immunohistochemical staining of PKM2 is more convenient for early diagnosis. Therefore, methylation signatures were not a convenient tool for the early detection and predicting the prognosis in LUICA. So, lack of immunohistochemical stain of PKM2 should be listed as one of limitation of this study.
3. P256-257, ---, Importantly, 257 nine abnormal methylationsites, including—
Suggestion
A space is needed between the methylation and site
---, Importantly, 257 nine abnormal methylation sites, including—

---

## Round 0.3 · accepted · Accept

Many thanks for your outstanding manuscript. We are delighted to inform you that your above manuscript has been reviewed and accepted for publication in PeerJ. Congratulations!